# Mechanistic Illustration: How Newly-Formed Blood Vessels Stopped by the Mineral Blocks of Bone Substitutes Can Be Avoided by Using Innovative Combined Therapeutics

**DOI:** 10.3390/biomedicines9080952

**Published:** 2021-08-03

**Authors:** Fabien Bornert, François Clauss, Guoqiang Hua, Ysia Idoux-Gillet, Laetitia Keller, Gabriel Fernandez De Grado, Damien Offner, Rana Smaida, Quentin Wagner, Florence Fioretti, Sabine Kuchler-Bopp, Georg Schulz, Wolfgang Wenzel, Luca Gentile, Laurent Risser, Bert Müller, Olivier Huck, Nadia Benkirane-Jessel

**Affiliations:** 1INSERM (French National Institute of Health and Medical Research) UMR 1260, Regenerative Nanomedicine, CRBS, 1 Rue Eugène Boeckel, 67000 Strasbourg, France; bornertfabien@gmail.com (F.B.); francois_clauss@hotmail.com (F.C.); g.hua@unistra.fr (G.H.); yidouxgillet@unistra.fr (Y.I.-G.); laetitia.keller@icloud.com (L.K.); fernandezdegrado@unistra.fr (G.F.D.G.); damien.offner@hotmail.fr (D.O.); ranasmaida@hotmail.com (R.S.); wagner.quentin@gmail.com (Q.W.); fiorettioce@gmail.com (F.F.); kuchler@unistra.fr (S.K.-B.); mckind75@gmail.com (L.G.); huck.olivier@gmail.com (O.H.); 2Faculty of Dental Surgery, University of Strasbourg, University Hospital Strasbourg (HUS), 8 Rue de Sainte Elisabeth, 67000 Strasbourg, France; 3Department of Pediatric Dentistry, University Hospital Strasbourg (HUS), 1 Place de l’Hôpital, 67000 Strasbourg, France; 4Biomaterials Science Center, University of Basel, Gewerbestrasse 14, CH-4123 Allschwil, Switzerland; georg.schulz@unibas.ch (G.S.); bert.mueller@unibas.ch (B.M.); 5Institute of Nanotechnology, Karlsruhe Institute of Technology, Campus North, Building 640, DE-76131 Karlsruhe, Germany; wolfgang.wenzel@kit.edu; 6Toulouse Institute of Mathematics, UMR 5219 University of Toulouse, CNRS UPS IMT, 31062 Toulouse, France; laurent.risser@math.univ-toulouse.fr

**Keywords:** therapeutic bone filling, pro-angiogenic smart nanotechnology, hybrid bone substitute, smart nano-active complexes

## Abstract

One major limitation for the vascularization of bone substitutes used for filling is the presence of mineral blocks. The newly-formed blood vessels are stopped or have to circumvent the mineral blocks, resulting in inefficient delivery of oxygen and nutrients to the implant. This leads to necrosis within the implant and to poor engraftment of the bone substitute. The aim of the present study is to provide a bone substitute currently used in the clinic with suitably guided vascularization properties. This therapeutic hybrid bone filling, containing a mineral and a polymeric component, is fortified with pro-angiogenic smart nano-therapeutics that allow the release of angiogenic molecules. Our data showed that the improved vasculature within the implant promoted new bone formation and that the newly-formed bone swapped the mineral blocks of the bone substitutes much more efficiently than in non-functionalized bone substitutes. Therefore, we demonstrated that our therapeutic bone substitute is an advanced therapeutical medicinal product, with great potential to recuperate and guide vascularization that is stopped by mineral blocks, and can improve the regeneration of critical-sized bone defects. We have also elucidated the mechanism to understand how the newly-formed vessels can no longer encounter mineral blocks and pursue their course of vasculature, giving our advanced therapeutical bone filling great potential to be used in many applications, by combining filling and nano-regenerative medicine that currently fall short because of problems related to the lack of oxygen and nutrients.

## 1. Introduction

Traumas, tumors, and infections could all induce bone defects. Small, non-critical sized bone defects do not pose a significant challenge, and could be treated with conventional approaches [1]. Critical-sized bone defects, however, are generally considered impossible to treat by conventional medical treatments, and require either bone substitutes or bone grafts [2,3,4,5,6]. Although the size varies depending on the anatomic location of the defect and the state of the surrounding tissue, bone defects larger than 2.5 cm are considered critical-sized bone defects in humans [7,8].

Recently, biphasic biomimetic bone substitutes were developed to replace autologous bone transplants, which are currently the gold standard in orthopedic surgery and traumatology [9,10,11]. Mimicking the natural composition of the bone tissue, these biomaterials contain an organic phase (e.g., collagen) associated to a mineral phase (e.g., hydroxyapatite). The former offers suitable conditions for the cells to graft, proliferate, and differentiate, the latter fills the bone defects, replacing the bone in terms of resistance to mechanical stimuli [9,10,11,12,13]. This new generation of bone substitutes aims at overcoming important limitations of the autograft technique, as the necessity of a double surgery and, therefore, occupies a relevant place in the field of orthopedic surgery.

Regardless of the approach used, the treatment of large bone defects remains a challenge for medical practitioners. Large engineered cellular constructs fail to integrate into the host tissues, because the lack of nutrients and oxygen leads to cell necrosis or ischemia, especially at the core of the graft. This largely depends on the slow speed by which the host vasculature colonizes the implant, which implies that several weeks are needed for the functional vascularization of implants larger than a few millimeters [14,15,16,17,18,19,20].

It has been reported that, for long-bone defects larger than 6 cm, vascularized bone substitutes induce a faster healing than conventional ones; therefore, the ideal bone substitute should actively promote its own vascularization from the host tissues [18,19,21]. For bone tissue engineering, several types of bioactive materials were developed, which couple improved capabilities for tissue repair with the presence of angiogenic factors, like Vascular endothelial growth factor (VEGF) [22,23], platelet-derived growth factor (PDGF) [24,25], bone morphogenetic protein (BMP) [26], or fibroblast growth factor (FGF) [27,28]. Among these, VEGF was shown to play a key part in bone repair and to increase bone healing in vivo [29,30,31]. The 165-amino acid form of VEGF (VEGF165) has angiogenic activity and can bind heparin (HEP) through its heparin-binding domain [32,33,34,35]. However, both the kinetic of release of VEGF and its local concentration are crucial for the vascularization of biomaterials. A swift local release of highly concentrated VEGF promotes increased vascular permeability, leading to the formation of non-functional blood vessels [36]. Opposed to that, a sustained release of low doses of pro-angiogenic molecules is essential to support the correct vascularization of implanted biomaterials and boost their efficiency in tissue repair [37,38,39,40,41,42].

The advent of third generation biomaterials expanded the array of possibilities to achieve a functional engraftment of the implant, with the addition of living cells [43]. Human umbilical vein endothelial cells (HUVECs), for example, have been largely used for their capacity to generate vascular networks in vitro; however, owing to their limited accessibility, they are hardly transferable to the clinic [10,17]. Recently, adult mesenchymal stem cells (MSCs) were shown to improve tissue healing, thanks to their angiogenic, antiapoptotic and immunomodulatory effects [44,45,46,47,48]. Moreover, it was shown that mesenchymal stem cells also play a role in the maintenance of a long-lasting vasculature in engineered constructs, as they can differentiate into both endothelial and perivascular cells [21,49,50,51].

The main objective of this work was not to regenerate bone, we only focused on critical bone lesions, which could not be regenerated through regenerative medicine or filled with the usual bone substitutes due to the lack of vascularization. Therefore, we developed a strategy to induce the functional vascularization of large bone substitutes, in vivo, using angiogenic factors and materials already used as bone substitutes. We used (1) a biphasic bone substitute currently used in the clinic; (2) our patent-pending smart nano-active complex (SNC) technology, providing the release of growth factors; (3) a complex of VEGF165 and HEP, to induce vasculoneogenesis; and (4) multipotent human MSCs. Here we report how the vasculoneogenesis induced within the transplanted bone substitute by the pro-angiogenic SNCs promote a faster and more functional regeneration of the bone, improving the therapeutic potential of our bioengineered bone substitute in treating critical-sized bone defects.

## 2. Materials and Methods

### 2.1. Deposition of the HEP/VEGF165 Complexes

Drops of HEP (500 μg mL^−1^), VEGF165 (200 μg mL^−1^), or HEP/VEGF165 (500 μg mL^−1^/200 μg mL^−1^) in 20 mM/0.15 mM Tris/NaCl, pH 6.8 solution (Sigma-Aldrich, Saint-Quentin-Fallavier, France) were laid on cover glass and dried. Salt crystals from buffer were solubilized in deionized water, by 2 rinsing steps of 5 min each. This deposition procedure was repeated 6 times to increase the quantity of material.

### 2.2. Scanning Electron Microscopy (SEM) and Atomic Force Microscopy (AFM) Study

In order to analyze the formation of HEP/VEGF-SNCs on the bone substitutes, samples were fixed with 4% paraformaldehyde (PFA) for 10 min at 4 °C. After dehydration, the specimens were observed by mean of SEM (either with Hitachi TM1000 or FEG Sirion XL; FEI) in conventional high vacuum mode with a secondary electron detector. A commercial stand-alone AFM microscope Solver Pro (Nt-Mdt Inc., Moscow, Russia) was used to acquire AFM images. Tapping imaging mode was used, with NSG 10 cantilever with a typical resonance frequency of 105 kHz, and a spring constant of 2 N m^−1^. The image resolution was set to 512 *×* 512, with a scanning rate of 1 Hz. Images were analyzed using the open source software Gwyddion 2.24 [52].

### 2.3. Smart Nano-Active Complexes Deposition on Bone Substitute

Antartik^®^ sponges (10% collagen I and III, 90% ceramic; Medical Biomat, Vaulx-en-Velin, France) were cut in 5 mm wide fragments and placed in a 96-well plate. They were sterilized with UV light (254 nm, 30 W, distance 20 cm, 30 min exposure). Chitosan/HEP/VEGF-SNCs were applied via layer-by-layer deposition, as previously described [38]. Briefly, bone substitutes were alternately dipped in 500-µg mL^−1^ chitosan solution (Protasan UP CL 113, NovaMatrix, Sandvika, Norway) and 500-µg mL^−1^/200 ng mL^−1^ HEP/VEGF complex solution in 20 mM/0.15 mM Tris/NaCl, pH 6.8. After each bath, bone substitutes were rinsed three times for 5 min in Tris/NaCl buffer. Before use, bone substitutes were equilibrated in serum-free Dulbecco’s Modified Eagle Medium (D-MEM).

### 2.4. Molecular Modeling of the HEP-VEGF165 Interactions

To prepare the starting structure for simulation, the HEP Sodium salt (CAS 9041-08-1) and VEGF165 heparin binding domain coordinates (PDB id: 2VGH, 55 amino acids) were extracted from the PDB structure files [53]. Partial atomic charges for heparin sodium salt molecule were assigned based on the AM1-BCC method using the antechamber function of AmberTools [54,55]. The van der Waals and bonded parameters for HEP were taken from the general amber force field (GAFF) [56]; the AMBER format files of these molecules were then converted to the GROMACS format using the ACPYPE python script. Using ClusPro and PatchDock docking programs, complexes of heparin and VEGF were generated [57,58,59]. The coordinates of the 6 best complexes were the selected to generate the topology file for each complex to run molecular dynamics (MD) simulations.

MD simulations were performed using GROMACS-4.6 for a period of 30 ns using explicit water model. The complex was placed in the center of a cubic periodic box and solvated by the addition of SPC water molecules. The net charge on the system was then neutralized by adding counter ions as required. The energy minimization was done using the steepest descent algorithm. The temperature and pressure were maintained at 300 K and 1 atm using the v-rescale temperature and Parrinello–Rahman pressure coupling method [60,61]. Production simulations were performed for 30 ns with a 2 fs time step. In order to calculate the interaction free energies of the heparin and VEGF complexes, we used the MM/PBSA protocol. The calculations were performed using the g_mmpbsa tool82, which implements the MM-PBSA approach using the GROMACS software packages [62]. The MM/PBSA energies were obtained from samples of 100 snapshots (for each complex) that were extracted from the MD trajectories. All calculations were done using computational time on BWUniCluster Karlsruhe.

### 2.5. Cell Culture

Green fluorescent protein-expressing HUVECs (PELOBiotech, Martinsried, Germany) and hMSCs from the bone marrow (PromoCell, Heidelberg, Germany) were cultured in the respective complete media (endothelial growth medium; mesenchymal stem cells growth medium, PromoCell) at 37 °C in a humidified atmosphere with 5% CO_2_. Moreover, 1.5 × 10^4^ hMSCs were seeded on each Antartik^®^ sponge fragment deposited or not with HEP/VEGF-SNCs, and cultured for 7 days in mesenchymal stem cell growth medium. After 7 days, 5.5 × 10^4^ GFP-HUVECs were seeded on the same samples and cultured in a medium consisting of half hMSCs growth medium, and half endothelial growth medium [63]. Bone substitute fragments seeded with cells were cultured for a total of 21 days, and then processed according to downstream applications.

### 2.6. Cell Biocompatibility Analysis

alamarBlue^®^ (Thermo Fisher Scientific, Waltham, MA, USA) was used to assess cell metabolic activity over time (*n* = 4). At 3, 14, and 21 days after GFP-HUVECs seeding, cells were incubated in 10% alamarBlue^®^ in D-MEM without phenol red (Lonza, Levallois-Perret, France) at 37 °C and 5% CO_2_. After 4 h, the supernatant was transferred to 96-well plates and the absorbances at 570 and 595 nm were measured in a Multiskan FC plate reader (Thermo Fisher Scientific, Illkirch-Graffenstaden, France) to determine the percentage of reduced alamarBlue^®^.

### 2.7. Immunofluorescence and Immunohistochemistry

Bone substitutes seeded with cells were fixed with 4% PFA for 10 min at 4 °C and rinsed with PBS, then demineralized and embedded in paraffin for 4 µm serial sections. Rabbit anti-human PECAM1/CD31 (Microm, Brignais, France) and non-species-specific anti-PECAM1/CD31 (Abcam, Paris, France), then secondary antibody raised against rabbit antibodies and coupled with Alexa 594 fluorochrome (Molecular Probes, Life Technologies, Thermo Fisher Scientific, Illkirch-Graffenstaden, France), diluted 1:200 in PBS 1% BSA, were used. After multiple rinses in PBS, samples were incubated 10 min in 200 nM 4′,6-diamidino-2-phenylindole (DAPI; Sigma-Aldrich, Saint-Quentin-Fallavier, France), for nuclear counterstaining. Specimens were observed with a LEICA DM4000B epifluorescence microscope (Leica Microsystems, Nanterre, France).

Mouse calvarial samples were fixed overnight at 4 °C in 4% paraformaldehyde and embedded in paraffin. Sections (10µm) were used for immunohistochemistry (IHC) staining of osteopontin (Opn) in the osteoblasts. Slides were then blocked with 12% of Avidin (in PBS) serum (Vector Laboratories, Burlingame, CA, USA) and incubated with monoclonal anti-mouse Opn primary antibody (dilution 1:200) (for IHC of Opn, Santa Cruz Biotechnology, #SC 21742, Clinisciences, Nanterre, France) at 4 °C, followed by incubation with biotinylated anti-rabbit/mouse IgG (Vector Laboratories, Eurobio Scientific, Les Ulis, France) for 30 min at room temperature. The staining was visualized using the ImmPact^TM^ Dab Detection Kit (Vector, #SK4105, Vector Laboratories, Eurobio Scientific, Les Ulis, France) by treating the sections for 30 min at room temperature. The slides were mounted using Aqueous base mounting medium (ImmunoLogic, #VWRK4058, VWR, Strasbourg, France).

### 2.8. Bone Substitute Implants in Nude Mice

The experimental protocol fulfilled the authorization of the “Ministère de l’Enseignement Supérieur et de la Recherche” under the agreement numbers APAFIS#7740-2016112116448550 (20 September 2016) and APAFIS#26926-2020080410295545 (20 December 2020). The Ethics Committee of Strasbourg named “Comité Régional d'Ethique en Matière d'Expérimentation Animale de Strasbourg (CREMEAS)” specifically approved this study. Nude male mice (Crl: NIH-Foxn1nu; Charles River, L’Arbresle, France), 6 weeks old, were anesthetized with an intra-peritoneal injection of 100 mg kg^−1^ of ketamine (VIRBAC Santé Animale; Centravet, Nancy, France) mixed with 10 mg kg^−1^ of Xylazine (Rompun^®^ 2%, VIRBAC Santé Animale; Centravet, Nancy, France), and placed on a heating plate kept at 37 °C. Five days before implant, each bone substitute was seeded with 1.0 × 10^5^ hMSCs. For subcutaneous implant, a dorsal skin incision was performed and the bone substitute (diameter = 5 mm) was placed between the skin and the muscle below. Incisions were sutured with resorbable material and mice were kept under observation for the whole experimentation time. After either 4 (*n* = 6) or 12 (*n* = 12) days post-implant (dpi), implanted mice were sacrificed with an intra-peritoneal injection of a lethal dose of ketamine. For implants in bone critical size defect, skin incision was performed, calvaria (parietal zone of the skull) was drilled using a sterile round burr (500 μm deep and 5 mm in diameter) and bone substitute (diameter = 5 mm) was placed in the defect. Incisions were sutured with resorbable material and mice were kept under observation for the whole experimentation time. Mice (*n* = 12) were sacrificed 12 dpi and explants were subject to histological, TEM and/or micro-CT assessments.

### 2.9. Histological Staining

For HE staining, subcutaneous explants were fixed with 4% PFA, demineralized, and embedded in paraffin for 7-µm serial sectioning. Samples were then subject to HE staining and observed on a Leica DM4000B microscope (Leica Microsystems, Nanterre, France). For quantitative analysis of the vascularization, ImageJ software was used. Number, mean size, and total area of the blood vessels found in the explants were analyzed. At least 5 images per section, 4 sections per sample were analyzed.

For Mallory staining, the explants were fixed with Bouin Hollande solution for 2 days. Then, they were dehydrated through a series of increasing ethanol concentrations, cleared with toluene, and embedded in paraffin wax. Sections were cut at 10 μm using a sledge microtome and mounted on glass slides. After the removal of paraffin wax, the sections were stained using Mallory staining for 2 days.

For Gomori staining, the explants were fixed with Bouin Hollande solution for 2 days and embedded in paraffin wax. Sections were cut at 10 μm using a sledge microtome and mounted on glass slides. After the removal of paraffin wax, the sections were stained using Gomori trichrome.

### 2.10. Micro-Angiography and Quantitative Analysis of the Vasculature

Twelve days after subcutaneous implantation, mice were subject to deep general anesthesia (sodium Pentobarbital 120 mg Kg^−1^). After opening the thoracic cage, an infusion needle was placed in the left ventricle. Mice were in turn perfused (rate of 2 mL min^−1^) with heparin (50 U mL^−1^; to purge the cardiovascular system), then with 4% PFA, PBS, and finally with radiopaque silicone rubber (Microfil^®^ MV-122, Flow Tech Inc. South Windsor, CT, USA). After perfusion, the heart was clamped to avoid leaks of the contrast agent; mice were then placed at 4 °C, overnight to allow the polymerization of the contrast agent. Explants were then post-fixed in 4% PFA for 48 h and then demineralized with Ethylenediaminetetraacetic acid (EDTA, 15%, pH = 7.4) for 1 week at 37 °C under constant slow agitation. The explants were mounted in 1% agar, in order to avoid any movement of the sample during micro-CT acquisition. The tomography experiments were carried out using the micro-CT X-ray system nanotom^®^ M (GE Sensing & Inspection Technologies GmbH, Wunstorf, Germany) equipped with a 180 kV–15 W high-power nanofocus tube with a tungsten transmission target. The X-ray micro-CT was performed with an isotropic pixel size of 5 μm^2^ and a field of view of about 15.4 × 12.0 mm^2^. For each measurement, the sample was irradiated by X-rays of 60 kV acceleration voltage and 310 mA beam current. At each rotation angle position, six images were acquired and averaged to a projection. Moreover, 1700 projections over 360° resulted in a total scan duration of about 100 min.

The data acquisition and reconstruction were performed with the phoenix datos|x 2.0 software (phoenix|x-ray, GE Sensing & Inspection Technologies GmbH, Wunstorf, Germany).

Grey level images obtained from micro-CT scan were segmented into two classes to distinguish the voxels within the vessels from those on the outside. For each image, manually tuned threshold was applied to intensity levels. A structuring element 10 μm wide was applied to each image in order to de-noise the segmentation. Then, the connected sets of voxels smaller than 20 μm^3^ were erased. After segmentation, the resulting vessels were sketched as skeletons, where a skeleton is defined as the ordered set of the points that defines both the center line and the local radius of a vessel-like shape, as for the algorithm previously described [64].

### 2.11. Transmission Electron Microscopy (TEM)

Explants at 12 dpi or 4 wpi (weeks post-implant) were fixed in 2.5% glutaraldehyde and 2.5% PFA in 0.1 M cacodylate buffer, pH 7.4 after decalcification. The samples were post-fixed in 1% osmium tetroxide, dehydrated, conditioned in propylene oxide, and embedded in Epon 812, Spurr epoxy resin (Electron Microscopy Sciences, Ft. Washington, PA). Semithin sections (2 µm) were stained with 1% Toluidine blue in 1% sodium borate, examined by Leica optical microscope (LEICA DMLB, Leica Microsystems GmbH; Germany). Ultrathin sections (70 nm) were prepared on a Leica Ultracut UCT ultra microtome (Leica Microsystems), contrasted with uranyl acetate (Laurylab, Brindas, France) and lead citrate (Euromedex EMS, Souffelweyersheim, France) and examined at 70 kV with a Morgagni 268D electron microscope (FEI-Phillips, Thermo Fisher Scientific, Illkirch-Graffenstaden, France). Digital images were captured using a Mega View III camera (Soft Imaging System, Münster, Germany).

### 2.12. Statistical Analysis

Statistical analysis was carried out with BioStatGV (Sentiweb, France) and Prism5 (GraphPad, La Jolla, CA, USA). The alamarBlue metabolic assay, histological data, and microangiography data were analyzed with unpaired, two-tailed Student’s *t*-test. When variance was found different between sets of data, then the Welch correction was applied.

## 3. Results

### 3.1. Design and Modeling of the Pro-Angiogenic Smart Nano-Active Complexes

In order to improve the vascularization of a graft implanted in a critical-sized bone defect, we nano-functionalized an FDA-approved biphasic bone substitute with pro-angiogenic nano-active complexes (patent pending) (Figure 1A,B) [38,39,40,41]. Heparin is critical for the angiogenic activity of VEGF, increasing both the kinase activation of VEGF receptor 1 and the mitogenic activity of VEGF [34,35,53,65]. Therefore, we first investigated the suitability of using the HEP/VEGF_165_ complex to functionalize the bone substitute. Heparin, VEGF_165_, or a complex thereof was deposited dropwise on a glass coverslip, and visualized by scanning electron microscopy (Figure 1B and Appendix A). Complexed HEP/VEGF_165_ formed large agglomerates (60.0 ± 15.4 nm), as confirmed by AFM imaging (Appendix A).

To understand why complexed HEP/VEGF aggregates had such a large size, we modeled the molecular interaction between HEP and VEGF_165_ using molecular dynamics simulations (Figure 1C). Heparin is a linear sulfated polysaccharide, with high negative charge, while the surface electrostatic potential of the HEP-binding domain of VEGF_165_ is positively charged [66]. Docking simulations of the HEP model to the NMR model of VEGF_165_ were therefore performed using the sulfate anions of HEP and the arginine (Arg) residues of VEGF_165_ as a guide to model the molecular interactions. Most of the positive Arg residues are either clustered in the central domain of VEGF_165_ (Arg35, Arg39, Arg46, Arg49) or within a loop in its N-terminus (Arg13, Arg14). These regions, according to the calculated electrostatic potential of the solvent accessible surface, represent the binding site for a high negatively charged molecule, such as HEP. The HEP/VEGF_165_ complex models generated using molecular docking were subjected to steepest decent energy minimization, followed by equilibration and a 30 ns molecular dynamics production run. The complex models were then ranked according to their energy, by means of the Molecular Mechanics Poisson–Boltzmann Surface Area (MM/PBSA) method [67]. The resulting HEP/VEGF_165_ complex structure with the lowest energy (Figure 1C) clearly indicates that the net charge plays a significant role in the affinity of HEP (stick model in Figure 1C) to VEGF_165_ (homology model in Figure 1C). These results can be explained by the fact that the sulfate groups of HEP (yellow in Figure 1C) strongly interact with the side chains of the Arg residues and the leucine 17 and threonine 47 residues of VEGF_165_, stabilizing the HEP/VEGF_165_ complex and allowing the formation of large agglomerates.

### 3.2. Bone Substitutes Equipped with HEP/VEGF-SNCs Improve In Vitro Organization of Endothelial Cells

The pro-angiogenic HEP/VEGF_165_ complexes (HEP/VEGF, in short) were integrated into chitosan SNCs and deposited on the bone substitute by using the nano-reservoir technology. The nano-active bone substitutes were observed at the SEM and compared to non-functionalized, chitosan-only SNCs (NF-SNCs). After six cycles of deposition, we observed a homogeneous distribution of either HEP/VEGF- or NF-SNCs, on both the mineral (Figure 2A,C) and the collagen (Figure 2B,D) portions of the bone substitute. To investigate the pro-angiogenic effects of the nano-functionalized bone substitute, either HEP/VEGF-SNCs, HEP/bovine serum albumin (BSA)-SNCs or NF-SNCs were deposited on fragments of bone substitutes that were in turn seeded with hMSCs (at t_0_) and Green fluorescent protein-expressing human umbilical vein endothelial cells (GFP-HUVECs, at day 7). The organization of the endothelial cells was monitored after 21 days of culture. In the presence of HEP/VEGF-SNCs, GFP-HUVECs organized themselves in vessel-like structures with visible lumen (red asterisks in Figure 2E). On the contrary, in the presence of either NF-SNCs or HEP/BSA-SNCs (Figure 2E), the endothelial cells remained mostly distributed as single cells on the graft. The pro-angiogenic activity of the SNCs was assessed by counting the number of GFP-HUVECs organized in vessel-like structures. On functionalized therapeutic bone substitute, these were on average 3.4 ± 0.5 cells long, while on bone substitute control or on functionalized with HEP/BSA, these were 1.2 ± 0.1 and 1.4 ± 0.2 cells long, respectively (*p* ≤ 0.001) (Figure 2F). Since the presence of HEP/BSA-SNCs produced results similar to those found for NF-SNCs, we concluded that the capacity of the nano-active bone substitute to promote the organization of endothelial cells in vitro depended on the presence and the cellular availability of the HEP/VEGF complex. We also assessed the metabolic activity of the cells seeded on non-functionalized bone substitute and compared it with that of cells seeded in the presence of HEP/VEGF-SNCs, by means of alamarBlue assay. In both conditions, the reduction of the alamarBlue increased from day 0 to day 21 (Figure 2G), suggesting that the nano-functionalized bone substitute was not cytotoxic. No significant differences were observed between NF- and HEP/VEGF-SNCs along the culture period considered. However, while the cells cultured with NF-SNCs showed an abrupt metabolic increase between day 3 and day 14 (*p* ≤ 0.1), the cells cultured in the presence of HEP/VEGF-SNCs showed a steady metabolic increment (*p* ≤ 0.05 and *p* ≤ 0.01, for day 3–14 and day 14–21, respectively). Since metabolic increases are generally associated with cell proliferation and growth, our data suggest that HUVECs proliferate less in the presence of strong differentiation cues, like those provided by the HEP/VEGF-SNCs.

### 3.3. In Vivo Vasculoneogenesis Induced after Subcutaneous Implantation of Nano-Active Bone Substitute

Bone substitutes functionalized with HEP/VEGF-SNCs improved the organization of endothelial cells in vitro, eliciting the formation of vessel-like structures. Therefore, we assessed if and how the angiogenic nano-active bone substitute could effectively trigger vasculoneogenesis in vivo. We subcutaneously implanted either HEP/VEGF-SNCs or NF-SNCs bone substitutes in the dorsal of nude mice. Prior to implant, the bone substitutes were seeded with hMSCs and cultured for 7 days. Bone substitutes were explanted from mice after 4 or 12 dpi, and their degree of vascularization was evaluated at the histological levels (Figure 3A,B). In general, explanted nano-active bone substitutes showed a better recruitment of host blood vessels compared to NF ones (yellow arrowheads in Figure 3A,B). Neither inflammatory infiltration, foreign body granulomas, scarring incidents, nor signs of rejection were observed at the implantation site. The quantification of the blood vessels found in the core of the bone substitutes showed a significant increase only in the presence of HEP/VEGF-SNCs (14.8 ± 1.7 mm^−1^ and 51.0 ± 12.9 mm^−1^ at 4 and 21 dpi, respectively; *p* = 0.002), compared to NF bone substitutes (13.2 ± 2.74 mm^−1^ and 22.6 ± 8.0 mm^−1^ at 4 and 12 dpi, respectively) (Figure 3C, left panel). No significant differences were observed at 4 dpi in the diameter of the blood vessels recruited (17 ± 1 µm vs. 18 ± 2 µm, in HEP/VEGF-SNCs and NF-SNCs bone substitutes, respectively) (Figure 3C, central panel) and in the relative vessel area (below 1% in either condition) (Figure 3C, right panel). The picture changed quite dramatically at 12 dpi, when both the size and the relative area of the host blood vessels found in the HEP/VEGF-SNCs bone substitutes increased 1.5- (*p* = 0.002) and 8-fold (*p* = 0.0003), respectively, while no differences were observed in the NF-SNCs bone substitutes (Figure 3C). These results show that the presence of HEP/VEGF SNCs increased the number and the size of the blood vessels recruited from the host tissues in the core of the bone substitute, a condition necessary to support the functional engraftment of the implant and to unleash its therapeutic potential.

### 3.4. Human MSCs Seeded on the Bone Substitute Actively Contributed to Vasculoneogenesis

The vessels observed within the bone substitutes subcutaneously implanted in nude mice look well connected to the surrounding vessels because none of the implanted animals suffered from bleeding. In order to have a closer look at the morphology of the newly-formed vessels, we analyzed them at the ultrastructural level. The vessels found within the bone substitutes were characterized by the presence of tightly connected endothelial cells of normal morphology, surrounded by mural cells that provide structural stability to the vessel (Figure 3D,E). The functionality of the blood vessels that are found within the bone substitute after implant, together with the speed at which they formed, are both of crucial importance to avoid the necrosis of the cellular component of the graft. Both functionalized and non-functionalized bone substitutes were seeded with hMSCs before implant. Mesenchymal stem cells are multipotent stem cells known to differentiate in many cell types, like osteoblasts, adipocytes, chondrocytes and even neurons [68]. Several studies also showed that MSCs could efficiently differentiate in endothelial cells in vitro [50,51]. Therefore, we investigated whether the hMSCs seeded on the bone substitute contributed to the formation of the blood vessels found within the grafts subcutaneously implanted. By means of immunohistochemistry, we detected several human platelet and endothelial cell adhesion molecule 1 (PECAM1)-positive endothelial cells (Figure 3F, top panels) which took part to the formation of the new blood vessels in the bone substitute (white asterisks in Figure 3F). As shown in the immunohistochemistry using an antibody that cross-reacted with both human and mouse PECAM1, these cells were a substantial portion of the endothelial cells found within the graft (Figure 3F, mid panels), suggesting a non-trivial contribution by the transplanted hMSCs. As expected, no signal was detected using the anti-human-PECAM1 antibody on a control mouse bone (Figure 3F, lower panels). These results indicate that the formation of new blood vessels within the implanted bone substitutes resulted from the simultaneous presence of both the pro-angiogenic SNCs and the hMSCs. We therefore concluded that the release of angiogenic growth factors and the differentiation of the transplanted hMSCs in endothelial cells synergistically contributed to the vasculoneogenesis observed within the nano-functionalized bone substitutes.

### 3.5. Pro-Angiogenic Bone Substitutes Promoted the Formation of New Bone in a Critical-Sized Calvarial Bone Defect

To investigate if the nano-functionalized bone substitutes could improve the speed of new bone formation, a portion of the skull roof 5 mm large and 0.5 mm deep was drilled out in nude mice (induced critical-sized calvarial bone defect), and filled with either HEP/VEGF-SNCs or the NF-SNCs bone substitutes (see Appendix A). Moreover, four wpi, mice were sacrificed and macroscopic observation of NF-SNC bone substitutes revealed a clear boundary between the implant and the healthy bone and a patchy appearance of the implant, which reflected the hybrid composition of the bone substitute (*n* = 5; Figure 4A). In contrast, the boundaries between the HEP/VEGF-SNC bone substitutes and the healthy bone were not evident and the appearance of the implant looked indistinguishable from the healthy bone, suggesting a very good integration of the newly-formed bone with the surrounding bone (*n* = 5; Figure 4A). Histological analysis with either Gomori or Mallory staining confirmed the better integration of the HEP/VEGF-SNC bone substitutes, suggesting the formation of abundant new bone, which engulfed the mineral portions of the bone substitute. On the contrary, the new bone formation in NF-SNC bone substitutes looked patchy, with large areas of intact mineral bone substitute in between (Figure 4B). Immunohistochemistry against Osteopontin confirmed the abundance of newly-formed bone in HEP/VEGF-SNC bone substitute (Figure 4C). Toluidine blue staining of semi-thin sections confirmed the continuous presence of both capillaries and osteocytes within the HEP/VEGF-SNC bone substitutes, where capillaries within the NF-SNC bone substitutes were only found between the mineral portions of the implanted bone substitute (Figure 4D). Transmission electron microscopy analysis of ultrathin sections showed the presence of osteoblasts and osteocytes within the HEP/VEGF-SNC bone substitute, but not within the NF bone substitute (Figure 4D). Together, these data showed that the abundant vascularization induced by the HEP/VEGF-SNC bone substitute accelerated the formation of new bone after the critical-sized defect induced in the mouse calvarial bone, compared to the NF bone substitute.

### 3.6. Nano-Active Bone Substitutes Promoted Vasculoneogenesis in a Critical-Sized Calvarial Bone Defect

We then tested the pro-angiogenic effects of the nano-active HEP/VEGF bone substitutes in the critical-sized bone defect mouse model. At 12 dpi, implanted animals were perfused with a radiodense rubber contrast agent and the bone substitutes were in turn explanted and subject to micro-CT scan. Relevantly, we could not detect any leakage of the contrast agent in the surrounding bone of none of the implanted mice (Appendix A), suggesting that the newly-formed vessels were both functional and perfectly integrated in the host vascular system. The 3D micro-angiographies of the transplanted mice (3D rendering of micro-CT scan images; Supplementary movie) were prepared in order to show the influence elicited by the presence of the HEP/VEGF-SNCs on the activation of vasculoneogenesis within the graft (Appendix A, black arrowheads). Differently to what we observed with the empty SNCs (Figure 5A,B, left panels), the newly-formed vessels penetrated intimately the HEP/VEGF-SNCs deposited bone substitutes, virtually spanning the entire volume of the implant (Figure 5A,B, right panels). Remarkably, when we used Standard Euler distances to generate distance maps to the closest neighboring vessel from the segmented micro-CT images acquired, we saw that both the trend of the curves and the mode of the distributions (0.32, 0.25, and 0.05 mm for 1NF, 2NF, and 3NF, respectively; 0.06, 0.02 and 0.05 mm for 1F, 2F and 3F, respectively) (Figure 5C) clearly indicated that the average distance of any point to the nearest neighboring vessel is smaller in the nano-active bone substitutes compared to the NF ones (*p* = 0.0423). The analyses also revealed that the vascular density in NF bone substitutes was 0.75 ± 0.99%, where in HEP/VEGF-SNCs bone substitutes was 2.8 ± 0.76% (*p* = 0.0167), which is almost four times higher.

Additionally, the diameter of the blood vessels found in the implanted grafts was measured. In general, bone substitutes deposited with HEP/VEGF-SNCs were colonized by vessels of a larger size (Figure 5D). While the large majority of the vessels found in non-nano-functionalized bone substitutes had a diameter smaller than 0.01 mm (93.55% vs. 84.49% in HEP/VEGF- and NF-SNCs bone substitutes, respectively; *p* = 0.091), blood vessels with a diameter between 0.02 and 0.03 mm were almost three times more frequent in HEP/VEGF-SNCs bone substitutes compared to NF-SNCs ones (1.91% vs. 0.68% in HEP/VEGF- and NF-SNCs bone substitutes, respectively; *p* = 0.0011). Moreover, vessels larger than 0.05 mm could only be found in HEP/VEGF-SNCs bone substitutes (Figure 5D).

In summary, these data indicate that the presence of the HEP/VEGF-SNCs entrapped into nano complexes combining heparin and chitosan as a coating of bone substitute mineral blocks promoted the formation of a denser cloud of vessels with larger diameters within the treated bone defect, an essential precondition for the successful filling of the defective bone.

## 4. Discussion

The speed at which the host vasculature colonizes implanted 3D biomaterials is approximately 10 μm per day [16,17,18,20]. Consequently, grafts implanted in critical-sized bone defects take several weeks for the proper vascularization, leading to necrosis, especially at the core of the implant, and to a failure to properly engraft [43]. In order to promote vasculoneogenesis inside bone substitutes, angiogenic growth factors, such as VEGF, incorporated into the scaffold material were reported to be promptly released upon the degradation of the scaffold [31,37,69,70]. In the case of bone substitutes, however, rapid degradation of the implant is not advisable, as the scaffold material has to fill the bone defect and compensate for the loss of its mechanical properties. On the other hand, a passive release of angiogenic factors, such as that found in the Augment^®^ Bone Graft system (Wright Focus Excellence, USA), soaked in 0.3 mg/mL recombinant human platelet-derived growth factor B (rhPDGF-BB), also leads to problems related to both the initial high concentration and then quick depletion of the growth factor [36,43]. Therefore, an accessible, localized, and sustained physiological concentration of VEGF is required for mature blood vessels to successfully develop within the bone defect.

The strategy presented in this work is based on the nano-functionalization of bone substitute material, which allows the release of therapeutic angiogenic molecules. In our approach, VEGF_165_ was complexed with HEP in order to maximize its angiogenic effects. The evidence we collected revealed that the HEP/VEGF_165_ complex formed larger aggregates than either VEGF_165_ or HEP alone (Appendix A), as a result of the chemical–physical interaction with HEP, as anticipated with our computer modeling (Figure 1C). The functionalization of bone substitutes using smart nano-active complexes (SNCs) is effective in overcoming the side effects associated with a high local dose of VEGF. Moreover, free VEGF_165_ has a very short half-life when exposed to the extracellular environment (approximately 90 min), yet does not degrade within the chitosan SNCs, and is therefore available to the cells for a prolonged time, as we and others have shown in previous studies [22,29,41]. In this work, we showed how the HEP/VEGF-SNCs induced the organization of endothelial cells in vessel-like structures in vitro (Figure 2E,F). This was also proved in vivo. Either implanted subcutaneously (Figure 3A–C) or in critical-sized calvarial bone defects (Figure 5A–D), the pro-angiogenic smart bone substitutes were able to promote vasculoneogenesis. Although the spatial distribution of the host blood vessels in one of the NF-SNC bone substitutes implanted in the calvarial defect was found to be similar to that of the HEP/VEGF-SNC bone substitutes, its vascular density was lower (1.9%) compared to the average found in the nano-active bone substitutes (2.8%). The most striking difference between NF- and HEP/VEGF-bone substitutes was, however, in the size of the newly-formed blood vessels. This could be explained as a difference in the maturity of the vessels, as observed in the histological specimens. Most importantly, the higher density and the bigger diameter of the newly-formed vascular network observed in HEP/VEGF-SNCs bone substitutes, promoted a better regeneration of the bone within critical-sized defects in mouse calvarial bone (Figure 4A–D). Following the implant of mineral-based bone substitutes, one major drawback is the long persistency of the mineral blocks, which prevents the growth of new vasculature within the bone substitute and slows down the replacement by newly-formed bone. In our HEP/VEGF hybrid bone substitutes, the new blood vessels meet fewer mineral blocks that are in their way and are thus able better to support the formation of new bones (Figure 5 and Figure 6; NF: non-functionalized HEP/VEGF-SNCs, F: functionalized HEP/VEGF-SNCs). Most importantly, we have observed the newly-formed vessels penetrated intimately the HEP/VEGF-SNCs deposited bone substitutes, virtually spanning the entire volume of the implant, which is totally different from what we observed with the empty SNCs (Figure 5A,B). After the microCT quantitative and qualitative analyses for the new blood vessels formed with or without SNCs, we elucidated the mechanism to understand how the new-formed vessels can no longer encounter mineral blocks and pursue their course of vasculature (Figure 6).

With the emergence of third-generation biomaterials, approaches using MSCs as therapeutic agents have garnered much interest for clinical applications [44,45]. These cells are not only able to differentiate into mature tissues, but could also modulate the immune response, prevent apoptosis, and promote angiogenesis via the secretion of trophic-factors [71]. Moreover, MSCs have been shown to stabilize newly-formed blood vessels by differentiating into both endothelial and mural cells [49,51]. The combined use of hMSCs, endothelial cells, and 3D biomaterials was shown to increase the formation of both vascular networks and new bone tissue [41,42,71]. Based on this knowledge, we opted to combine the pro-angiogenic bone substitute with MSCs, assessing how these two elements could synergistically promote vasculoneogenesis in vivo. As shown here, besides the recruitment of endothelial cells from the host, the hMSCs pre-seeded on the nano-functionalized bone substitute actively contributed to the formation of new blood vessels, which is demonstrated by the several endothelial cells of human origin found in the vasculature within the bone substitute (Figure 3F).

In summary, our results clearly indicate that the presence of the HEP/VEGF-SNCs entrapped into nano complexes as a coating of bone substitute mineral blocks promoted the vascularization stopped before by the mineral component. We demonstrated that our therapeutic bone substitute could indeed offer an effective substitution for either auto- or allografts in the treatment of large bone defects. Concomitantly, our innovative strategy allows modifications in the composition of the active elements (both cells and growth factors) so that it could be used for other tissues application.

## 5. Conclusions

Presently, autologous bone grafting (auto-transplant) is the gold standard in the treatment of large bone defects. With ≥2 million procedures per year worldwide, it is the second most common tissue transplantation after blood transfusion [11]. However, it has downsides. Bone autografts present more complications than the use of synthetic bone substitutes in terms of infections; moreover, they have a higher cost [72,73]. Allogenic bone grafts are even more expensive than autografts, as they have to be properly treated prior to the clinical use [74,75]. However, autografts necessarily introduce a second operative site, a longer operating room time, and a worse and longer-lasting post-operative chronic pain, which increase both the distress of the patient and the overall costs of the technique [64]. Therefore, advanced therapeutics materials are needed to overcome the current limitations (i.e., the lack of blood vessels recruitment from host tissues and the necrosis induced at the core of the implanted bone substitute) and to induce the vascularization of the implanted bone substitutes, which in turn favors the regeneration of functional bone tissue. In this study, we developed a therapeutic bone substitute based on the combined presence of proangiogenic, smart nano-active complexes that aim to improve vasculoneogenesis in critical-sized bone defect. The results presented in this work have great relevance from both biomedical and public health perspectives. They showed that our innovative strategy, applied to a bone substitute already used in the clinic, was able to (i) induce the functional organization of endothelial cells in vessel-like structures in vitro, and (ii) promote vasculoneogenesis in bone substitutes implanted either subcutaneously or in critical-sized calvarial bone defects.

To conclude, we reported here the quantitative analysis of the functional vasculature formed in the hybrid bone substitutes implanted in critical-sized bone defects. We have also elucidated the mechanism to understand how the newly-formed vessels can no longer encounter mineral blocks and pursue their course of vasculature, giving to our advanced therapeutic bone filling’s great potential to be used in many applications, combining filling and regenerative medicine that currently fall short because of problems related to the lack of oxygen and nutrients.

## Figures and Tables

**Figure 1 biomedicines-09-00952-f001:**
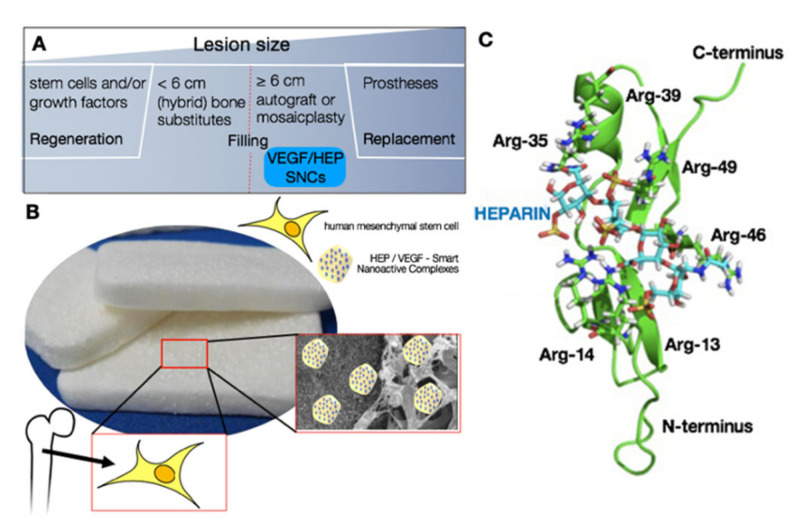
Third-generation hybrid bone substitute nano-functionalized with pro-angiogenic molecules. (**A**) Current strategies for bone repair. (**B**) The combination of pro-angiogenic smart nano-active complexes (SNCs), human mesenchymal stem cells, and biphasic bone substitute used in this study. (**C**) Molecular modeling of the HEP/VEGF complex, where VEGF is displayed as homology model (green), and HEP as stick model (blue). Sulfur moieties are shown in yellow.

**Figure 2 biomedicines-09-00952-f002:**
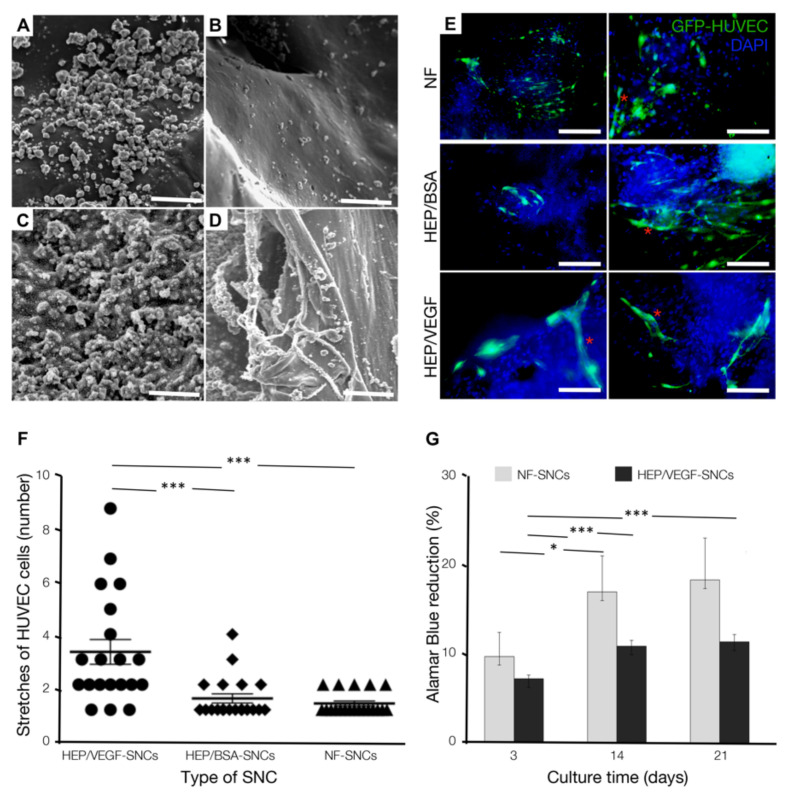
Nano-functionalization of the bone substitute with HEP/VEGF-SNCs and its effect on the organization of umbilical cord-derived endothelial cells in vitro. (**A**–**D**) SEM pictures of bone substitutes deposited with either - (NF-) (**A**,**B**) or HEP/VEGF- (**C**,**D**) SNCs. The SNCs were found on both the mineral (**A**,**C**) and the protein (**B**,**D**) constituents of the biphasic bone substitute. Scale bar in (**A**,**C**): 5 μm; in (**B**,**D**): 10 μm. (**E**) Fluorescence micrographs of GFP-HUVECs co-seeded with hMSCs on the bone substitute deposited with either NF-, HEP/BSA-, or HEP/VEGF-SNCs, after 21 days of culture. Nuclei counterstained with DAPI. Scale bars: 150 µm. (**F**) Number of GFP-HUVECs found organized in vessel-like structures (red asterisk in (**E**)) on NF-, HEP/BSA-, or HEP/VEGF-SNCs bone substitutes. Bars represent mean ± SEM (*n* = 20 per condition); ***: *p* < 0.001. (**G**) The presence of reduced alamarBlue^®^ (indicating increased cell proliferation/cell metabolism) was quantified for the cells seeded on bone substitutes either deposited with NF- or HEP/VEGF-SNCs, after 3, 14, and 21 days of culture. Values expressed as mean ± SEM (*n* = 4). *: *p* < 0.1; ***: *p* < 0.01.

**Figure 3 biomedicines-09-00952-f003:**
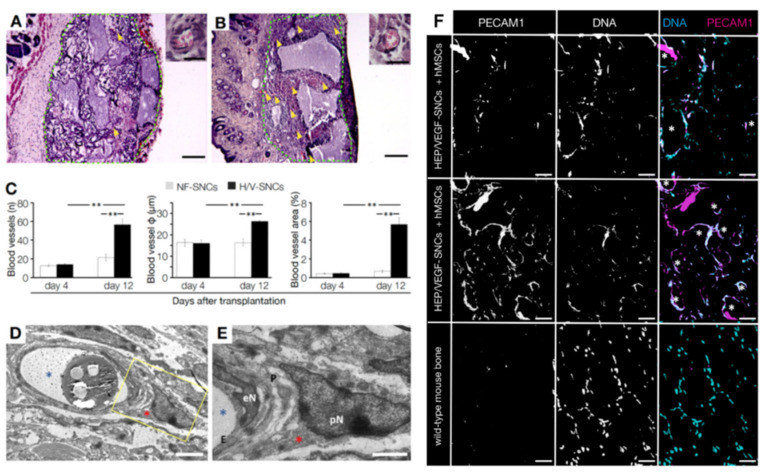
Vasculoneogenesis in bone substitutes subcutaneously implanted in nude mice. (**A**,**B**) H&E staining of bone substitutes (green lines) with either empty (NF-) (**A**) or HEP/VEGF- (**B**) SNCs, at 12 dpi. Yellow arrowheads indicate blood vessels. Scale bars: 200 µm. The images at the top right show enlargements of blood vessels. Scale bars: 20 µm. (**C**) Quantitative analysis of the vessels found in the implanted bone substitutes. The average number and diameter of the vessels found, together with the average surface covered are given at 4 and 12 dpi. Values are expressed as mean ± SD of at least five images/section and four sections/sample. **: *p* ≤ 0.05. (**D**,**E**) Ultrastructural view of a blood vessel found in the HEP/VEGF-SNCs bone substitute, shown as a transverse section. Mural cells enveloping the blood vessels (e.g., the pericyte indicated with a red asterisk) were found in the nano-active bone substitute. (**E**) This is the enlargement of the area delimited by the yellow frame in (**D**). No blood cells are in the lumen of the vessel, as a contrast agent was perfused (residues indicated with a blue asterisk). E: endothelial cell; eN: nucleus of the endothelial cell; P: pericyte; pN: nucleus of the pericyte. Scale bars: 5 µm in d, 2 µm in e. (**F**) Endothelial cells of human origin in the bone substitute, as revealed with anti-human (top panels) or anti-human/mouse PECAM1 antibody (mid panel). Mouse control bone was negative to the anti-human PECAM1 antibody (lower panel). Scale bars: 200 µm.

**Figure 4 biomedicines-09-00952-f004:**
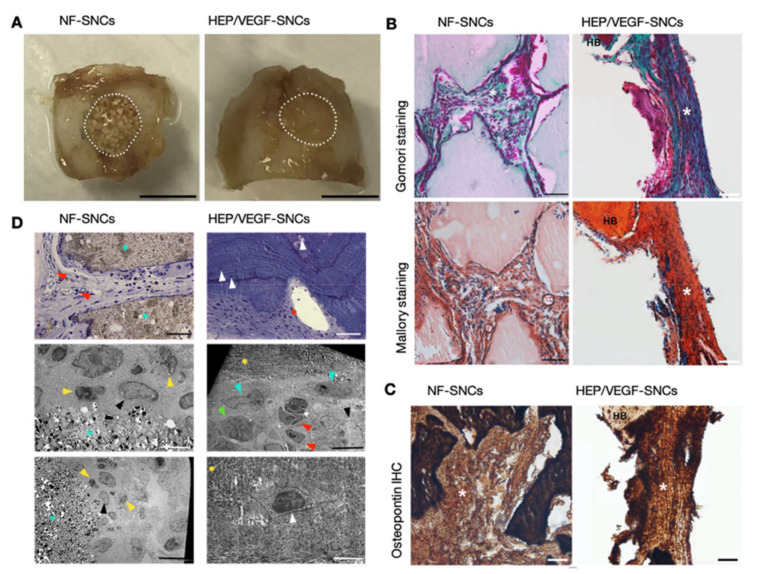
Bone regeneration in critical-sized calvarial bone defects implanted with bone substitutes. (**A**) Macroscopic evaluation of the calvarial bones at 4 wpi implanted with either NF- or HEP/VEGF-SNCs. Scale bar: 5 mm. (**B**) Gomori and Mallory stains of sections of the calvarial bones implanted with either NF- or HEP/VEGF-SNCs. In Gomori trichrome stain, collagen is light blue, erythrocytes are pink, osteoblasts/osteocytes are violet/brown; in Mallory trichrome stain, cartilage is blue, bone components are orange to red and nuclei are brown. HB: Healthy bone; white asterisk: newly-formed bone within the implanted bone substitute. Scale bar: 50 µm. (**C**) Sections of the specimens shown in (**B**) underwent immunohistochemistry analyses against Osteopontin. HB: healthy bone. Scale bar: 50 µm. (**D**) Toluidine blue staining on semi-thin sections (upper panels) and transmission electron micrographs on ultrathin sections (mid and lower panels). Red arrowhead: capillary; white arrowhead: osteocyte; cyan arrowhead: osteoblast; yellow arrowhead: granulocyte; black arrowhead: megakaryocyte; green arrowhead: erythroblast; red asterisk: capillary lumen with blood cells; cyan asterisk: mineral part of the biphasic bone substitute; yellow asterisk: bone matrix. Scale bar: 25 µm (upper panels); 10 µm (mid panels), 5 µm (lower panels).

**Figure 5 biomedicines-09-00952-f005:**
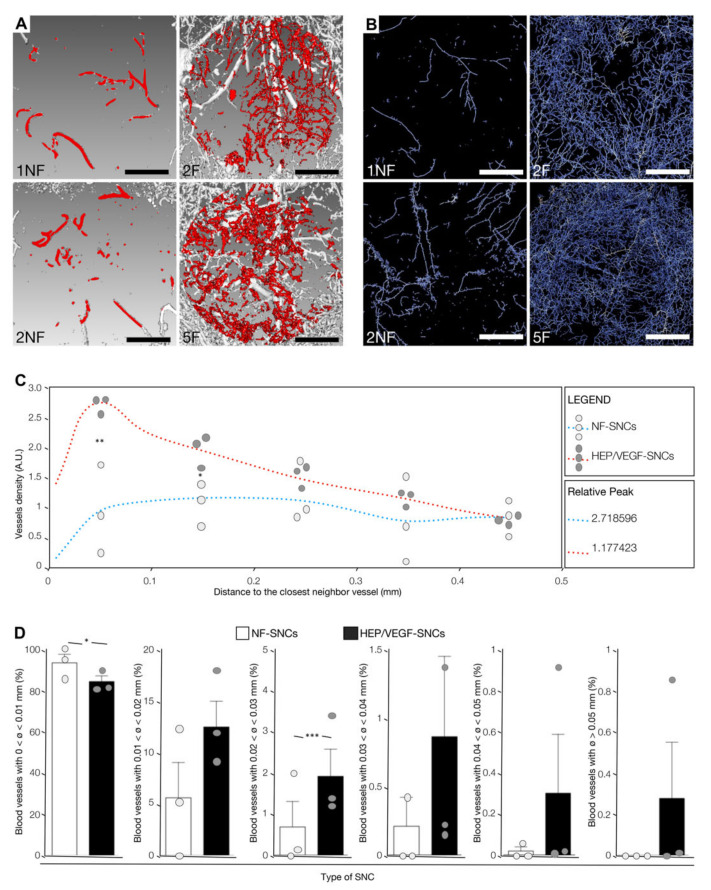
Quantitative analysis of the vasculature found in the bone substitutes implanted in critical-sized calvarial bone defects. (**A**) Micro-CT scans of four representative bone substitutes, deposited with empty (1NF, 2NF) or pro-angiogenic HEP/VEGF (2F, 5F) smart nano-active complexes (SNCs), 12 days after implant. Vessels within the bone substitutes are shown in red. Scale bars: 1 mm. (**B**) Skeletons of the segmented CT images of the vascular networks found in 4 representative bone substitutes, either deposited with NF-SNCs (1NF, 2NF) or with HEP/VEGF-SNCs (2F, 5F). Lighter colors display vessels of larger size. Scale bars: 1 mm. (**C**) Relative density of the vascular networks in 6 bone substitutes, either deposited with NF-SNCs (light grey dots) or with HEP/VEGF-SNCs (dark grey dots), represented in function of the average distance to the closest neighbor vessel. The red and blue dotted lines represent the mean values for HEP/VEGF-SNC and NF-SNC deposited bone substitutes, respectively. *: *p* < 0.1; **: *p* < 0.05. (**D**) Hybrid bar/dot plot charts of the relative number of vessels in NF-SNC or HEP/VEGF-SNC deposited bone substitutes, as a function of the blood vessel diameter, 12 days after implant. *: *p* < 0.1; ***: *p* < 0.01. Bars represent Mean ± SD (*n* = 3).

**Figure 6 biomedicines-09-00952-f006:**
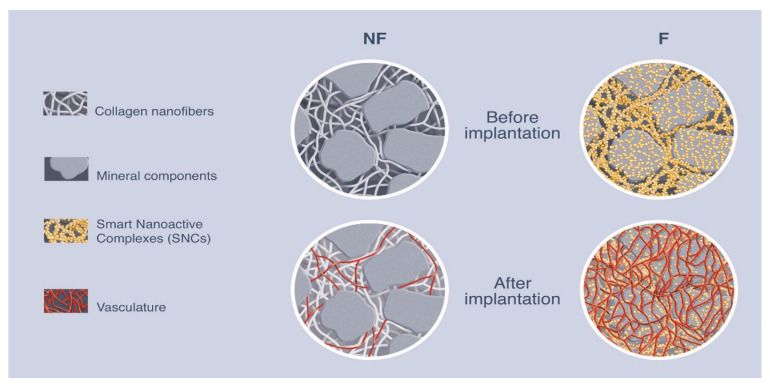
Mechanistic illustration of vascularized new generation of hybrid bone substitute.

## Data Availability

Data is contained within the article or supplementary materials.

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
