# Peer review of "Mechanistic Illustration: How Newly-Formed Blood Vessels Stopped by the Mineral Blocks of Bone Substitutes Can Be Avoided by Using Innovative Combined Therapeutics"

_biomedicines, 2021, doi:10.3390/biomedicines9080952_

Round 1
Reviewer 1 Report
This well-written and well-presented manuscript is experimentally rich and the results and conclusion reported are globally convincing, except a few points that should be clarified or modified: 1. Refs are numerous, but also quite old overall. 2. The last paragraph of the introduction section is a bit confusing, both assuming that "the objective of the work was not to regenerate bone" and promising to demonstrate a "more functional regeneration of bone". 3. The scaffolds were seeded with hMSCs first, and only 7 days later, with HUVECs. The authors do not explain the reasons of this protocol. This should be discussed somewhere. 4. The description of the HEP/VEGF complexes by AFM and SEM is not really convincing, and in my sense, does not prove much. From the given protocol, the dimension of 60 nm derived from AFM stands for a dehydrated and thus, collapsed deposit, such that agglomerates in solution (from which they are intended to adsorb onto the substrates) may be potentially still much bigger. Do the authors think that such big agglomerates are really eligible components for LbL deposition? Nanosizer measurements of the agglomerates in suspension might help characterizing the complex suspensions. 5. Also, how the authors relate the molecular modeling of the complex with the formation of large agglomerates is not clear at all. 6. Why did the authors perform 6 cycles of layer-by-layer deposition, not less, not more? Are they sure that the layer-by-layer construction was effective. Considering the size of the agglomerates, there is a risk that the process even stopped after the first cycle. 7. Section 3.3: The diameters of the blood vessels indicated in the text do not correspond to the values reported in Figure 3C. 8. Section 3.3: The authors assumes that HEP/VEGF is "slowly released in physiological concentrations", but there are no evidences of a release and the concentrations in the data. 9. Introduction, section 3.4 and discussion: The authors assume a "cell contact-dependant sustained release" of the GF, but there are no data supporting both the release and its mechanism. The manuscript should be less assertive about these points. 10. The histological analyses and interpretations (Figures 3.3 and 3.4) are not always obvious, and many components are not distinguishable so easily. 11. I was not able to watch the movie in Supplementary Material (no access) 12. Discussion: the authors assume that "VEGF was complexed with HEP in order to maximize its angiogenic effects". This does not fit with the whole manuscript, where it appears that complexation was rather intended to transport VEGF with a negatively-charged polymer (HEP) in order to implement a layer-by-layer protocol in association with positively-charged chitosan. Typos. 1. 2.4 Molecular Modeling: "were assigned" instead of "was assigned" 2. Powers of ten should appear in superscript (eg. in 2.5 Cell Culture) 3. Figure 3: "Day 12" should appear on 1 line (twice), and "SNC" insteat of "SMC"
Author Response
This well-written and well-presented manuscript is experimentally rich and the results and conclusion reported are globally convincing, except a few points that should be clarified or modified:
- Refs are numerous, but also quite old overall.
Answers: We have added several new references.
- The last paragraph of the introduction section is a bit confusing, both assuming that "the objective of the work was not to regenerate bone" and promising to demonstrate a "more functional regeneration of bone".
Answers: We thank the reviewer for this important remark. Indeed, as indicated in the last paragraph of the introduction, the critical bone lesions could not be regenerated through regenerative medicine or filled with the usual bone substitutes due to the lack of vascularization. Thus, the main objective of this work was not to regenerate critical bone lesions, but to induce the functional vascularization of large bone substitutes. We have demonstrated in the Figures 4 and 5 that thanks to the induced vasculoneogenesis in a critical size calvarial bone defect, functionalized HEP/VEGF-SNF bone substitute promoted the formation of new bone as compared to the non-functionalized SNCs bone substitute.
- The scaffolds were seeded with hMSCs first, and only 7 days later, with HUVECs. The authors do not explain the reasons of this protocol. This should be discussed somewhere.
Answers: We thank the reviewer for the critical reading of our experimental protocols. It is well-known that MSC cells have the proangiogenic features (Boomsma et al., Plos one, 2012 (Ref. 47); Tao et al., Stem Cells Int. 2016 (new Ref. 48)). hMSCS were seeded firstly to the membrane, this procedure permitted the MSCs to secret the extracellular matrix (ECM) of MSC and proangiogenic factors, which could create the optimal condition for the adherence and growth of the HUVEC cells. In order to promote vessel formation, in the literature we found that HUVEC cells perform better when seeded some days after hMSCs, e.g. 14 days, like in Ren et al., Biomed Res Int. 2014 (Reference 63); however, after this time, hMSCs form multiple layers, therefore we halved the time to 7 days.
- The description of the HEP/VEGF complexes by AFM and SEM is not really convincing, and in my sense, does not prove much. From the given protocol, the dimension of 60 nm derived from AFM stands for a dehydrated and thus, collapsed deposit, such that agglomerates in solution (from which they are intended to adsorb onto the substrates) may be potentially still much bigger. Do the authors think that such big agglomerates are really eligible components for LbL deposition? Nanosizer measurements of the agglomerates in suspension might help characterizing the complex suspensions.
Answers: We thank the reviewer for this comment. To answer your question, we agree that such agglomerates (60nm) are eligible components for LbL deposition, Recently, we and others have reported the use of nanoparticles biggest than in this study (Wagner et al., Future medicine, 2016). In our opinion Nanosizer measurements of the agglomerates in suspension might not help for this kind of application because we will combine this complex with another component during the deposit and that will really change the agglomerates at the end.(Wagner et al., Nanomedicine (Lond), 2016).
- Also, how the authors relate the molecular modeling of the complex with the formation of large agglomerates is not clear at all.
Answers: We have changed the last sentence of the section 3.1.
- Why did the authors perform 6 cycles of layer-by-layer deposition, not less, not more? Are they sure that the layer-by-layer construction was effective. Considering the size of the agglomerates, there is a risk that the process even stopped after the first cycle.
Answers: Our laboratory has published several papers using patented Nanoreservoir technology (Mendoza-Palomares et al., ACS Nano, 2012; Eap et al., Biomed Mater Eng, 2012; Eap et al., Int. J. Nanomedicine, 2015; Keller et al., Nature Communications, 2019). We have demonstrated the effective step-by-step buildup of the active nanoreservoirs achieved by simple alternate immersions into a positively charged solution (for example; chitosan) and the negatively charged therapeutic agent solution from one to six cycles. The buildup of the nanoreservoirs embedding therapeutic agent was followed by quartz crystal microbalance (QCM-D), revealing linear growth in the wet adsorbed mass and the hydrodynamic thickness, and confirming the nanometric scale of the nanoreservoirs. No difference was observed with membranes nano-functionalized more than 6 cycles.
- Section 3.3: The diameters of the blood vessels indicated in the text do not correspond to the values reported in Figure 3C.
Answers: We thank the reviewer for this critical reading. All the values presented in the Figure 3C were verified, and the mistakes were corrected.
- Section 3.3: The authors assumes that HEP/VEGF is "slowly released in physiological concentrations", but there are no evidences of a release and the concentrations in the data.
Answer: The sentence is probably misleading and belongs more to the conclusions. The fact that the HEP/VEGF is actually released slowly by the SNCs, at physiological concentration, has been previously shown by our group (Schiavi et al., 2015; Wagner et al., 2016). Since a swift release of growth factors at high concentrations does impact negatively the surrounding cells and tissues, we assumed that the observed vasculoneogenesis owed more to the slow release by SNC, than to the mere presence of VEGF. the sentence has been downplayed: "These results show that the presence of HEP/VEGF SNCs increased the number and the size of the blood vessels recruited from the host tissues in the core of the bone substitute, a condition necessary to support the functional engraftment of the implant and to unleash its therapeutic potential."
- Introduction, section 3.4 and discussion: The authors assume a "cell contact-dependant sustained release" of the GF, but there are no data supporting both the release and its mechanism. The manuscript should be less assertive about these points.
Answers: as mentioned above, these are not assumptions based on the data presented here, but evidences previously published by our group. However, we decided to delete "cell contact-dependant sustained release" of the GF in the text.
- The histological analyses and interpretations (Figures 3.3 and 3.4) are not always obvious, and many components are not distinguishable so easily.
Answers: We’ll upload the figures with high resolution.
- I was not able to watch the movie in Supplementary Material (no access)
Answers: We’ll reupload the supplementary movie.
- Discussion: the authors assume that "VEGF was complexed with HEP in order to maximize its angiogenic effects". This does not fit with the whole manuscript, where it appears that complexation was rather intended to transport VEGF with a negatively-charged polymer (HEP) in order to implement a layer-by-layer protocol in association with positively-charged chitosan.
Answers: We agree with the reviewer that HEP is negatively-charge polyer, and the HEP-VEGF complex is also negatively charged which could be associated with positively-charged chitosan through layer-by-layer protocol. In addition, VEGF-HEP forms a more stable complex than VEGF alone which could consequently increase the angiogenesis capacity.
Typos.
- 2.4 Molecular Modeling: "were assigned" instead of "was assigned"
- Powers of ten should appear in superscript (eg. in 2.5 Cell Culture)
- Figure 3: "Day 12" should appear on 1 line (twice), and "SNC" insteat of "SMC"
Answers: All the mistakes have been corrected.
Reviewer 2 Report
- As stated in the title that newly-formed blood vessels are stopped by mineral blocks, I couldn’t find any results or images supporting this evidence on stoppage of blood vessels. Also as stated stoppage of blood vessels must result in necrosis of tissue. Have you observed necrosis in such cases of mineral block testing without the smart nanoactive complex?
- In section 2.3 regarding Smart Nanoactive Complexes Deposition on Bone Substitute, the commercial bone substitutes i.e. Antartik® sponges are dipped in Chitosan/HEP/VEGF solutions. Hence this is a surface adsorption as the SNC are not deposited during bone substitute preparation. Hence could you comment on the release of VEGF & HEP? According to me, surface adsorption would result in burst release of factors and as a result prolonged angiogenesis in relation to bone regeneration may be questionable.
- Overall the enhancement of angiogenesis by VEGF-HEP complex is already well known and established. Hence what’s the novelty of the study?
- Include the magnifications and scale bars for all the SEM and AFM images including the supplementary figures.
- In Figure 6, what do NF and F denote?
- As depicted in Figure 6, even in the absence of SNC, angiogenesis is observed whereas in the presence of SNC, angiogenesis enhanced. But this is contrary to the title, which indicates stoppage of blood vessels.
Author Response
Comments and Suggestions for Authors
- As stated in the title that newly-formed blood vessels are stopped by mineral blocks, I couldn’t find any results or images supporting this evidence on stoppage of blood vessels. Also as stated stoppage of blood vessels must result in necrosis of tissue. Have you observed necrosis in such cases of mineral block testing without the smart nanoactive complex?
Answers: We have demonstrated in the Figure 5 that VEGF-HEP functionalized bone substitutes promoted the vasculogeonenesis. In the section 3.6., the results indicate that the presence of the HEP/VEGF-SNCs entrapped into nano complexes as a coating of bone substitute mineral blocks promoted the formation of a denser cloud of vessels with larger diameters within the treated bone defect, an essential precondition for the successful filling of the defective bone.
- In section 2.3 regarding Smart Nanoactive Complexes Deposition on Bone Substitute, the commercial bone substitutes i.e. Antartik® sponges are dipped in Chitosan/HEP/VEGF solutions. Hence this is a surface adsorption as the SNC are not deposited during bone substitute preparation. Hence could you comment on the release of VEGF & HEP? According to me, surface adsorption would result in burst release of factors and as a result prolonged angiogenesis in relation to bone regeneration may be questionable.
Answers: We thank the reviewer for this important remark about the cell contact-dependent active release of therapeutic agents by nanoreservoir which is a patented technology (PCT/EP2012/052976). The Smart Nanoactive Complexes (SNCs) are not only deposited to the surface, but homogenously integrated to the nanofibers of the Antartik spongesnot by this technology (Mendoza-Palomares et al., ACS Nano, 2012; Eap et al., Biomed Mater Eng, 2012; Eap et al., Int. J. Nanomedicine, 2015; Keller et al., Nature Communications, 2019).
- Overall the enhancement of angiogenesis by VEGF-HEP complex is already well known and established. Hence what’s the novelty of the study?
Answers: We thank the reviewer to point out the issue of novelty. We agree with the reviewer that VEGF-HEP complex is well known to enhance the angiogenesis. However, in this study, we have demonstrated that VEGF-HEP complex released in a cell contact-dependent active manner by patented nanoreservoir technology (PCT/EP2012/052976) could efficiently promote the angiogenesis both in vitro and in vivo.
Our laboratories have published that a very low, but sufficient amount of BMP2 (100 ng/cm2 = 10.000 times lower than that found in the Medtronic implant (Inductos Collagen membrane)) is able to induce bone regeneration without triggering inflammatory side effects (Mendoza-Palomares et al., ACS Nano, 2012; Eap et al., Biomed Mater Eng, 2012; Eap et al., Int. J. Nanomedicine, 2015; Keller et al., Nature Communications, 2019).
- Include the magnifications and scale bars for all the SEM and AFM images including the supplementary figures.
Answers: All the scale bars are indicated in the legends.
- In Figure 6, what do NF and F denote?
Answers: NF means non-functionalized HEP/VEGF-SNCs, F means functionalized HEP/VEGF-SNCs. We have added these definitions (see lines 526-527).
- As depicted in Figure 6, even in the absence of SNC, angiogenesis is observed whereas in the presence of SNC, angiogenesis enhanced. But this is contrary to the title, which indicates stoppage of blood vessels.
Answers: We agree with the reviewer that in the absence of SNC, angiogenesis is observed as depicted in Figure 6. However, these few blood vessels are all around the mineral blocks and could not across the obstacles as indicated in the title that angiogenesis could not be occurred where there are mineral blocks. This is the major problem of lack of vascularization when filing the critical bone defects with the usual bone substitutes. In this figure, it is clear that with our HEP-VEGF-SNCs, the angiogenesis has been enhanced as observed by the reviewer, the newly-formed vessels can no longer encounter mineral blocks and pursue their course of vasculature.
Round 2
Reviewer 2 Report
A suggestion would be that the authors can depict the stoppage of blood vessels in a usual bone substitute vs the nano functionalized substitute more clearly with closely magnified CT images, if possible.
As indicated in the answer to query 6, stating blood vessels cannot cross the obstacle, this can be projected with a magnified CT.